# Population Pharmacokinetics of Doxycycline, Administered Alone or with N-Acetylcysteine, in Chickens with Experimental *Mycoplasma gallisepticum* Infection

**DOI:** 10.3390/pharmaceutics14112440

**Published:** 2022-11-11

**Authors:** Tsvetelina Petkova, Antoaneta Yordanova, Aneliya Milanova

**Affiliations:** 1Department of Pharmacology, Animal Physiology, Biochemistry and Chemistry, Faculty of Veterinary Medicine, Trakia University, 6000 Stara Zagora, Bulgaria; 2Department of Social Medicine, Healthcare Management and Disaster Medicine, Faculty of Medicine, Trakia University, 6000 Stara Zagora, Bulgaria

**Keywords:** doxycycline, mycoplasmosis, N-acetylcysteine, population pharmacokinetics, poultry

## Abstract

Mycoplasmosis is a bacterial infection that significantly affects poultry production, and it is often controlled with antibiotics, including doxycycline. The conducted study aimed to determine population pharmacokinetic (PopPk) parameters of doxycycline in healthy (n = 12) and in *Mycoplasma gallisepticum*-challenged (n = 20) chickens after its oral administration via drinking water at the registered dose rate of 20 mg/kg b.w./24 h for five days, without or with co-administration of N-acetylcysteine (NAC, a dose of 100 mg/kg b.w./24 h) via the feed. Doxycycline concentrations in plasma were analyzed with the LC-MS/MS method. The values of *tvV/F* and *tvk_e_* were 4.73 L × kg^−1^ and 0.154 h^−1^, respectively, and they showed low BSV. A high BSV of 93.17% was calculated for the value of tlag of 0.8 h, which reflects the inter-individual differences in the water consumption. PTA was computed after Monte Carlo simulation with the registered dose for doxycycline. The target of %*f*T > MIC ≥ 80% and 100% can be achieved in 90% of the broiler population, after a correction for protein binding, for bacteria with MIC ≤ 0.5 mg × L^−1^ and 0.25 mg × L^−1^, respectively. The applied PopPk model did not reveal significant effect of *M. gallisepticum* infection and co-administration of NAC on pharmacokinetic parameters of doxycycline.

## 1. Introduction

*Mycoplasma* infections in poultry, very often caused by *Mycoplasma gallisepticum* and by *Mycoplasma synoviae,* are among the main reasons for significant economic losses in poultry farming [1,2,3]. *Mycoplasma gallisepticum* was described as a primary cause of chronic respiratory disease in chicken flocks [4,5]. *M. gallisepticum* can persist in the organism of chickens and cause chronic inflammation when the immune response is ineffective [6]. Moreover, co-infections by *M. gallisepticum* and other bacterial and viral pathogens lead to higher incidence of gross tissue lesions resulting in higher morbidity and mortality rates [6,7]. The worldwide experience of veterinarians has shown that, once it occurs in the poultry productive chain, mycoplasmosis is very difficult to eradicate [3]. Therefore, many strategies have been applied to control *Mycoplasma* infection. Treatment of the infected flocks with antibiotics, together with improved biosecurity and management practices and vaccination programs, are the three main measures devoted to eradication of mycoplasmosis [2]. However, application of antibiotics in farm animals is associated with frequent emergence of resistance in pathogenic bacteria, so strict regulations for their use in veterinary medicine have been introduced [8]. Tetracyclines are classified as antibiotics with alternatives in human medicine and are among the first-choice antibacterial drugs for the treatment of bacterial infections in animals [8].

Good activity of doxycycline, a tetracycline antibiotic, against *Mycoplasma* spp. has been described in the scientific literature [1,2,9]. Determination of minimum inhibitory concentrations (MIC) is a very important step in the rational use of antibiotics, but this information has to be used together with knowledge on drug pharmacokinetics. Doxycycline pharmacokinetics, after a single administration in poultry, has been reported [10,11,12]. There are data about its pharmacokinetics in broilers after oral administration with drinking water for five consecutive days [13]. In a recent pharmacokinetic-pharmacodynamic (Pk-Pd) study in healthy broilers, doxycycline showed excellent effectiveness against *M. gallisepticum* strain *S6* [14]. Most of the cited pharmacokinetic experiments were conducted in healthy animals, and pharmacokinetic analysis was performed after administration of doxycycline at a single dose. A single oral treatment of broilers with natural *M. gallisepticum* infection resulted in lower plasma concentrations, shorter elimination half-life and decreased values of total body clearance and volume of distribution [11]. There are insufficient data about doxycycline pharmacokinetics after repeated administration via drinking water in infected chickens. Recent literature data revealed that not only diseases can be a source of variability for orally administered antibiotics. Administration of the antibiotics via drinking water may contribute to extra variability in their pharmacokinetics, related to the differences in water consumption by individual animals [15]. Altogether the available data indicate that the knowledge on pharmacokinetics of doxycycline after administration via drinking water for five consecutive days in chickens with mycoplasmosis is insufficient. Population pharmacokinetics can be a powerful tool for refinement of rational use of doxycycline in chickens [16,17].

Together with the use of more advanced tools for pharmacokinetic studies and in the context of restricted use of antibiotics in farm animals, there are growing efforts for finding substances that can increase the susceptibility of pathogenic bacteria to antibacterial drugs. N-acetylcysteine (NAC) is a compound with antibacterial and antibiofilm activity, which can enhance the susceptibility to beta-lactam antibiotics [18,19]. NAC reduces the values of MIC of tetracycline for strains of *Staph. aureus*, *Strep. dysgalactiae* and *Strep. agalactiae*, except for that of *E. coli* [20]. It is often used in the treatment of chronic bronchitis because of its mucolytic activity and antioxidant properties [21]. A literature review suggest that NAC can be an important modulator of antibiotic activity and can have beneficial effect in inflammation of the respiratory system, but it does not provide information for pharmacokinetic interactions between doxycycline and NAC. Therefore, NAC was co-administered with doxycycline in chickens with respiratory infection caused by *M. gallisepticum*, and its effect on pharmacokinetics of the antibiotic was examined.

Based on the insufficient knowledge on doxycycline pharmacokinetics, the aim of the study was to use population pharmacokinetics for evaluation of the effect of *M. gallisepticum* infection and NAC co-administration on doxycycline pharmacokinetics after oral administration with drinking water for five consecutive days in broilers. Furthermore, Monte Carlo simulations were used for prediction of plasma concentration–time curves for a big population of chickens, and the probability of target attainment of T%> MIC ≥ 80 and 100% was assessed using published MIC values for *Mycoplasma* spp.

## 2. Materials and Methods

### 2.1. Drugs

HydroDoxx^®^ 500mg/g oral powder (Huvepharma NV, Antwerp, Belgium), containing doxycycline hyclate was used for administration with the drinking water. N-acetyl-L-cysteine (≥99%, Sigma-Aldrich, St. Louis, MO, USA) was applied in the feed of poultry. LC-MS/MS analysis of doxycycline plasma concentrations was performed by using the following reagents: doxycycline hyclate (HPLC grade ≥98%, Sigma-Aldrich Chemie GmbH, Taufkirchen, Germany), oxytetracycline hydrochloride used as internal standard (HPLC grade ≥ 95%, Sigma-Aldrich Chemie GmbH, Taufkirchen, Germany), trifluoroacetic acid (99.5%, Fisher Chemical, Fisher Scientific, Waltham, MA, USA), LC/MS grade acetonitrile OPTIMA^®^ (Fisher Chemical, Fisher Scientific, Waltham, MA, USA), formic acid for mass spectrometry ~98% (Honeywell Fluka™, Seelze, Germany) and water for chromatography (LC-MS grade, LiChrosolv^®^, Merck KGaA, Darmstadt, Germany).

### 2.2. Experimental Design

The poultry handling and experimental design complied with the Bulgarian legislation, and the investigation was conducted after receiving ethical approval from the Bulgarian Food Safety Agency (License 245/25.09.2019).

Fifty-two 1-day-old hybrid Ross chicks from both sexes were obtained from the commercial hatchery “Zhuliv” EOOD, Stara Zagora and were housed in the Biobase of the Faculty of Veterinary Medicine at Trakia University. One-day-old chicks (n = 36) were challenged with 1.10^8^ CFU/mL bacterial suspension of *M. gallisepticum* R low strain (Pendik Veterinary Control Institute, Istanbul, Turkey). Each chicken (n = 36) was injected with bacterial suspension in the left (0.1 mL) and in the right (0.1 mL) cranial thoracic air sac. During the first days after infection, a tendency to decreased appetite was observed. Infected broilers showed clinical signs of mycoplasmosis, such as nasal discharge, dyspnea and growth retardation. The morbidity rate was 44.3 ± 6.4%. Pathological examination of the dead chickens proved that the experimental inoculation resulted in successful induction of mycoplasmosis. Thickened serosa, fibrinous unilateral or bilateral airsacculitis in the thoracic and in the abdominal air sacs was observed. No clinical signs of the disease and mortality were observed in the healthy chickens during the experiment. The chickens were fed concentrated feed without addition of antibiotics and coccidiostats, according to their age (“Vladini Trading” EOOD, Chirpan, Bulgaria). Birds were monitored twice daily. They were kept in daylight for 18 h and in the dark for 6 h. Food and water were provided ad libitum. The feed consumption was recorded every day during the experiment. The detailed information about the feed composition and the reared conditions were described in the related paper about pharmacokinetics of N-acetylcysteine [22]. The pharmacokinetic studies started when broiler chickens were 40 days of age. Their body weight was registered one day before the treatment.

The chickens were allocated in four groups as described below:

The first group (n = 6 healthy chickens with mean body weight 2.79 ± 0.17 kg) was treated with doxycycline hyclate at a dose rate of 20 mg/kg b.w. for 24 h with the drinking water for five consecutive days. The water was freshly prepared every 12 h, and the consumed amount was registered. The recalculated real dose received by the broilers was 24.41 mg/kg b.w.

The second group (n = 6 healthy chickens with mean body weight 2.84 ± 0.29 kg) received doxycycline hyclate at a dose rate of 20 mg/kg b.w. every 24 h with the drinking water and N-acetylcysteine at a dose rate of 100 mg/kg b.w./24 h with the feed for five days. The actually received dose of doxycycline was 23.70 mg/kg b.w.

Broilers (n = 10, mean body weight of 2.47 ± 0.27 kg) from the third group were infected and treated orally with doxycycline hyclate via drinking water. The aimed dose was 20 mg/kg b.w. every 24 h, and the actually received dose of the antibiotic was 22.18 mg/kg b.w.

The chickens from the fourth group (n = 10, mean body weight of 2.48 ± 0.43 kg) were infected with *M. gallisepticum* and treated with doxycycline hyclate via drinking water at a dose rate of 20 mg/kg b.w./24 h. NAC was administered with food at a dose rate of 100 mg/kg b.w./24 h. The obtained dose of doxycycline was 23.78 mg/kg b.w.

Blood samples (0.8 mL) were obtained from *v. subcutanea ulnaris* before the treatment and at time 0 h and 0.5, 1, 2, 3, 4, 6, 9, 12, 14, 24, 120, 122, 124, 126, 132, 144, 152, 168 and 174 h. after the start of doxycycline treatment. Blood samples were obtained from six chickens from all experimental groups at every time interval. They were immediately placed in Eppendorf tubes containing heparin-sodium and were centrifuged at 850× *g* for 10 min. Plasma was transferred to new clean tubes and stored at −80 °C until analysis.

### 2.3. Determination of Plasma Concentrations by LC-MS/MS Analysis

Plasma concentrations of doxycycline were analyzed by LC-MS/MS method. Extraction of doxycycline from 500 µL of plasma samples was performed after addition of 10 µL internal standard (oxytetracycline, 5 µg × mL^−1^) and 65 µL trifluoroacetic acid. The precipitated samples were vortexed for 1 min and then centrifuged for 10 min at 10,800× *g* at 22 °C. The supernatant was carefully filtered through syringe filters with 0.22 µm pore size. The filtrate was transferred in LC-MS/MS vials, and 5 µL was injected into the LC-MS/MS system [23].

A Poroshell 120 EC C18 column (i.d. 4.6 mm, 100 mm, 2.7 µM, Agilent Technologies, Santa Clara, CA, USA) was used for chromatographic separation of tetracycline antibiotics. Gradient elution was performed with mobile phase A containing 0.1% formic acid in LC-MS grade water and mobile phase B containing 100% acetonitrile. The flow rate was 0.3 mL × min^−1^, and the run time was 12 min with a post-run of 5 min. The applied gradient elution program was as followed: 0–0.5 min (90% A, 10% B), 0.5–8 min (90% A, 10% B) and 8–12 min (2% A, 98% B). The LC-MS/MS system consisted of a 1260 Infinity II quaternary pump and a 1260 Infinity II Vial Sampler connected to a triple-quadrupole mass spectrometer Agilent 6460c with Agilent Jet Stream technology. The following conditions of MS/MS analysis were set: gas temperature (N_2_) 350 °C, 45 psi nebulizer gas N_2_, sheath gas N_2_ at 400 °C, sheath flow 12 L × min^−1^, capillary voltage of 4000 V, nozzle voltage of 500 V and dwell time of 200 ms. The qualification and quantification ions of doxycline were 445.1 m/z and 428.1 m/z and for the internal standard oxytetracycline values were 461.1 m/z and 444 m/z, respectively [23]. The internal standard was used at a final concentration of 0.1 µg × mL^−1^. The method was validated using plasma samples from untreated chickens at the following concentrations of doxycycline: 0, 0.005, 0.01, 0.05, 0.1, 0.25, 0.5, 0.75 and 1 µg × mL^−1^. The standard curve was linear (R^2^ = 0.991) between 0.01 and 1 µg × mL^−1^. The limit of detection was 0.02 µg × mL^−1^, and the limit of quantification was 0.08 µg × mL^−1^. The average accuracy was 98.25 ± 4.81%. The intra-day precision value was 2.55%, and the inter-day precision was 9.94%.

### 2.4. Determination of Plasma Protein Binding of Doxycycline in Chickens

The binding of doxycycline to plasma proteins was determined using Ultrafree—MC Centrifugal Filters with a hydrophilic PTFE membrane and 0.45 μm pore size (Merck KGaA, Darmstadt, Germany) according to the manufacturer’s instructions. The tests were performed in triplicate at five concentrations of doxycycline in plasma: 3, 2, 1, 0.5 and 0.1 μg × mL^−1^. Plasma samples (800 μL) were incubated for 1 h at 37 °C. After that, they were centrifuged first at 1000× *g* for 10 min, then at 2000× *g* for 20 min. Filtrate (5 μL) of each concentration was injected into the LC-MS/MS system. The concentration of doxycycline in plasma and ultrafiltrate was determined by the analytical method described above. The percentage of protein binding was calculated using the following formula:% protein binding = (CTP − CFP/CTP) × 100,(1)
where CTP is the total plasma concentration, and CFP is the unbound concentration in the filtrate [24].

### 2.5. Non-Compartmental Analysis

A non-compartmental analysis (Phoenix 8.3.4, Certara^®^, Cary, NC, USA) was used to calculate the pharmacokinetic parameters of doxycycline after multiple oral administrations with a dosing interval (tau) of 24 h in healthy broiler chickens and chickens experimentally infected with *M. gallisepticum*. The following pharmacokinetic parameters were calculated: C_max_, maximum plasma concentration; T_max_, time to reach C_max_; and AUC_0→∞_, area under the curve of plasma concentrations of doxycycline in the interval from 0 h to ∞. The extrapolated part of AUC was less than 2.16%, which was lower than the acceptable level for extrapolation of <20%. The data, including statistical analysis, are shown in the Appendix A.

### 2.6. Population Pharmacokinetic Analysis

Population pharmacokinetic modelling was performed with specialized software Phoenix 8.3.4 (Certara^®^, Cary, NC, USA). Population pharmacokinetic parameters were estimated with a first-order conditional estimator with extended least squares algorithm (FOCE ELS). First, several scenarios were evaluated on the basis of the results from healthy animals treated solely with doxycycline, for which a full data set was available for every chicken. After that the data of the orally treated poultry with doxycycline from all four groups were modelled simultaneously (12 healthy broilers with intense sampling, n = 19 samples/chicken; and 20 infected animals with less intense sampling, n = 9–10 samples/chicken during the entire experiment) and were used for final selection of the model and for the calculation of the population parameters. A two-compartmental model with first-order absorption and elimination was selected among the other evaluated models and, finally, used to fit the data. The parameterization was in terms of absorption rate constant (*k_a_*); volume of distribution (*V*); elimination rate constant (*k_e_*); distribution rate constants, which describe the distribution of the drug from the central to the peripheral compartment (*k*_12_) and from the peripheral to the central compartment (*k*_21_); and lag-time (tlag). The final population model was selected on the basis of the lowest values of Log Likelihood (-2LL), the Akaike Information Criterion (AIC), visual inspection of the plots, plots of visual predictive check (VPC) and the model comparer tool of Phoenix. The effect of co-variables, such as health status (healthy or *M. gallisepticum*-infected chickens) and drug administration (doxycycline alone or in combination with N-acetylcysteine), were tested for determination of the source of variability in the population. The tested co-variables did not improve the model and were excluded from the final population pharmacokinetic model. The model comparer tool of Phoenix and LRT test were used to discover the significant differences between the models with and without co-variables. The applied error model consisted of a multiplicative and additive error. A few concentrations (4.6% from the total number) that were below the limit of quantification were not included in the analysis.

An exponential model was used to estimate the between-subject variability (BSV) describing the biological variability. The pharmacokinetic parameters (*k_a_*, *V*, *k_e_*, *k*_12_, *k*_21_ and tlag) for each subject (*i*th animal) were determined with the following structural model:(2)$$V_{i}=tv_{V}\times \exp\left(\eta_{Vi}\right)$$
where *V_i_* is volume of distribution for an individual, *tv_V_* (also called θ) is the population value of *V* and η*_i_* (eta) is the deviation associated with each individual in the population. The values of *V* corresponded to *V*/*F* because of the orally administered drug. The other parameters were calculated with the same algorithm.

Normal distribution was assumed for η with mean 0 and a variance (*ω*^2^). Equation 3 was used to convert the variance to a coefficient of variation (*CV*%):(3)$$CV_{V}\left(\%\right)=100\times \sqrt{\left(\exp\left(\omega^{2}V\right)-1\right)}$$
where *ω*^2^*V* is the variance of the volume of distribution. The same algorithm was used for the other parameters.

The following equation was used to estimate the values of *Shrinkage*:(4)$$Shrincage=1-\frac{SD\left(\eta j\right)}{\omega j,j},$$
where *SD*(η_j_) is the standard deviation of the jth (observation) for all subjects, and *ω*_j,j_ is the estimate of the population variance of the jth random effect, j = 1, 2, …N, etc.

### 2.7. Simulation and Probability of Target Attainment Analysis

The established population pharmacokinetic model was used as a basis for generation of a virtual population by Monte Carlo Simulations in Phoenix 8.3.4. In total, 5000 Monte Carlo simulations were conducted to calculate estimates of % *f*T > MIC. The plasma concentration–time profiles were simulated for up to 24 h with increments of 1.5 h. The simulated curves were used for calculation of V and *k_e_*. Protein binding of doxycycline was set according to the results from the current investigation. The Probability of Target Attainment (PTA) was calculated for the applied dose, which was registered for treatment of bacterial diseases in poultry (20 mg/kg b.w. every 24 h) by using SPSS Statistics 26.0.0.1 for Windows (IBM, Armonk, NY, USA). The Pk-Pd breakpoint of %*f*T > MIC ≥ 80 and 100 was applied for doxycycline, which was used to predict the efficacy of tetracyclines [14]. A PTA of 90% was considered effective, taking into account the applied breakpoints. The times above MIC were computed for values of MIC of 0.25, 0.5, 1 and 2 µg × mL^−1^.

The following equation was used for estimation of T > MIC [25]:(5)$$\%T>\mathrm{MIC}=Ln\left(\frac{Dose\times f}{V\times \mathrm{MIC}}\right)\times \left(\frac{1}{ke}\right)\times \left(\frac{100}{\tau }\right),$$
where *Ln* is the natural logarithm, *f* is the fraction of unbound doxycycline, *V* is the volume of distribution, *k_e_* is the elimination rate constant and *τ* is the dosing interval. The correction for protein biding converts %*T* > MIC into %*f*T > MIC.

## 3. Results

The experimental infection with *M. gallisepticum* was successfully induced in groups of infected chickens, and the birds showed typical clinical signs of the disease. Data about FCR and water consumption are included in the Appendix A, Appendix A. Mycoplasmosis was proved by the observed pathological changes. Clinical signs of disease and mortality were not detected in the groups of healthy chickens.

Plasma concentrations of doxycycline after oral administration in healthy and *M. gallisepticum*-infected chickens were found as early as the first sampling interval, 0.5 h after the start of treatment (Appendix A from the Appendix A). Large variations in the values of doxycycline levels in plasma of individual broilers were observed at this time point, and few of them were close to the limit of detection (4.39% of the samples). At the last sampling point, 174 h after the start of the treatment, plasma concentrations of the antibiotic were determined in all broilers with the exception of one bird. The results from non-compartmental analysis did not reveal significant differences in the values of T_max_ and AUC_0→∞_ (Appendix A). The highest values of C_max_ were determined in the group of healthy broilers treated with doxycycline and NAC. This value was significantly higher than the values of C_max_ in the groups of the infected chickens, irrespective of the treatment. Plasma protein binding of doxycycline was found to be between 50.55 ± 12.35% for concentrations between 0.1 and 1 µg × mL^−1^ and 77.06 ± 7.5% for the higher concentrations of 2 and 3 µg × mL^−1^, respectively.

Population pharmacokinetic analysis was performed in order to find the source of differences between the experimental groups. Based on the plots of VPC and CWRES (conditional weighted residual values) versus TAD (time after dose, h), it was assumed that the selected population pharmacokinetic model adequately described the changes in doxycycline concentrations over time (Figure 1a,b). Evenly distributed data about zero, between y = −2 and y = 2, indicated no major bias in the structural model. The adequacy of the model was further verified from the plots of IPRED and PRED vs. observed concentrations (DV) of doxycycline (Figure 2a,b). Data were evenly distributed about the line of identity indicating that the structural model can properly describe changes of doxycycline concentrations for most individuals.

The typical values (tv) of the primary structural parameters and their standard errors (SE), coefficients of variation (*CV*%) and the 95% confidence intervals (CI) are presented in Table 1. Table 2 contains data about random components of the model, between-subject variability (BSV) and shrinkage. The value of *CV* for the typical parameters was <20%, indicating an adequate calculation of the population pharmacokinetic parameters. The value of *tvV/F* reflected the high distribution of doxycycline. Although there was an acceptable value for shrinkage, below 30% for tlag, high BSV was found (Table 2). The values of *tvk_a_* and *tvk_e_* indicated a different rate of absorption and elimination, respectively, with low degree of BSV. The values of shrinkage (>0.3) for *k_a_* and for the elimination rate constant *k_e_* (0.33) indicated that the model can be improved with a larger data set, especially characterizing the absorption phase. The values of shrinkage for the rest of the parameters were below 0.3, which allows us to accept that the model was not overparameterized. Clear bi-exponential curves were typical for the results of the groups of healthy animals when intense sampling design was applied. Co-variables, such as disease and co-administration of NAC, did not improve the prediction of the population data. These co-variates had no significant effect on the precision of calculation of population pharmacokinetic parameters and were removed from the model.

Further MCS were performed based on the selected population model, and the generated data were used for PTA analysis for four different MIC levels and considering the average percentage of unbound drug (0.50; percentage of protein binding 50.55 ± 12.35%). The results for *f*T > MIC were reported for a dosing regimen of 20 mg/kg b.w./24 h (Table 3). Ninety percent of the chickens had a *f*T > MIC of 79.60% (~80%) over the dosage interval of 24 h for a MIC of 0.5 mg × L^−1^, which corresponded to a total plasma concentration of 1 µg × mL^−1^.

## 4. Discussion

Refinement of use of antibiotics in veterinary medicine is a part of One Health approach to combat the emergence and spread of resistance among pathogenic bacteria [8]. New efforts are directed to increase the responsible and prudent use of antibiotics, especially in farm animals. Categorization of antibacterial drugs leads to limitation of the available antibiotic classes for treatment of animals, and tetracyclines are among the first-choice antibiotics. Doxycycline has been registered for control of mycoplasmosis in poultry for many years. Its pharmacokinetic properties were widely investigated, including in diseased animals, but often in well-controlled trials when poultry received the exact dose in the crops [10,11,26,27]. Usually, poultry flocks are treated via drinking water, and the uptake of the antibiotics is highly dependent on their behavior, diseased condition, body weight and other factors, which can be a prerequisite for significant inter-individual variations [28]. Apart from the necessity of basic knowledge on doxycycline pharmacokinetics in broilers obtained with classical experimental design, population pharmacokinetic experiments can be used to better understand the source of inter-individual variability [29]. The power of population pharmacokinetic analysis was demonstrated in the modelling of enrofloxacin pharmacokinetics in chickens with the drinking behavior of the treated animals [15].

The current experiments were planned to investigate doxycycline pharmacokinetics in broilers in conditions that are very close to those in poultry farms where mycoplasmosis is still a problem [30]. Therefore, the population approach was used to characterize doxycycline freely received with water in healthy and *M. gallisepticum*-infected chickens. The measured plasma concentrations were similar to the reported values in chickens with natural *M. gallisepticum* infection, and the values of C_max_ in the both groups of the infected broilers confirmed the published data on the achievement of lower concentrations in the diseased poultry [11]. The estimated pharmacokinetic parameters *tvV/F* and *tvk_e_* were in agreement with the published data of 5.29 L × kg^−1^ and 0.09–0.1 h^−1^, respectively [13,31]. One of the limitations of the population model was related to the calculated high values of shrinkage for the absorption rate constant *tvk_a_*. Taking into account the statistical components of the developed population model, it can be concluded that *tvk_a_* was calculated with certain bias. Its value was lower compared to the majority of published data (from 1.1 ± 0.24–0.83 ± 0.26 to 2.55 ± 1.40 h^−1^), which can be explained by the differences in the route of drug administration: via crop gavage or via drinking water in our experiments [11,31]. In a study with a similar design of drug administration but in younger broilers, a higher value of *k_a_* of 0.14 ± 0.10 h^−1^ was obtained [13]. The inclusion of lag time improved the fit of the data and properly reflected the time necessary for doxycycline to appear in systemic circulation following its uptake via drinking water. The values of shrinkage of *tvk_a_* indicated that collection of more data, especially at the absorption phase, can improve the model. It should be noted, however, that the population model revealed effect of variations in medicated water consumption on the absorption of the drug, which was reflected on the calculated variations in the lag time and the uncertainty of *tvk_a_*. The other tested models, including one-compartmental ones, also calculated values of shrinkage > 0.3 for some of the parameters (data not shown). The lack of improvement of the model with *M. gallisepticum* challenge as a co-variate suggests that experimental infection did not have an impact on pharmacokinetics of doxycycline. Typical values of the pharmacokinetic parameters were calculated based on the actually received doses by every group, which could contribute to the lack of differences between healthy and diseased animals.

NAC was administered to poultry as an antioxidant [32]. Recent experience shows that NAC can decrease the negative impact on the growth of chickens under heat and cold stress [33,34]. Based on the data about antioxidant and mucolytic activity, as well as discovered antibacterial properties of NAC, the other task was to evaluate its effect on pharmacokinetics of doxycycline in healthy and in *M. gallisepticum*-infected broilers by means of a population pharmacokinetic approach [35]. The conducted analysis did not reveal any effect of NAC co-administration on doxycycline pharmacokinetics. Published data by our group showed that NAC pharmacokinetics was not affected by doxycycline administration in broilers [22]. Altogether, these data suggest that pharmacokinetic drug–drug interactions between these compounds cannot be expected in broilers. The available registers for drug–drug interactions reveal that oxytetracycline and tetracycline should not be combined with NAC due to decreasing of the efficacy of acetylcysteine or incompatibility in a solution, respectively [36,37]. The lack of registered pharmacokinetic drug–drug interactions under the experimental conditions in this study is not sufficient to conclude that the combinations between doxycycline and NAC can be applied in treatment of respiratory infections. It is important to evaluate the effect on bacterial growth and replication. There is increasing evidence that NAC exhibits an antibacterial effect, although only at high concentrations [38]. In in vitro studies NAC demonstrated a high activity against carbapenem-resistant *Klebsiella pneumoniae* and *Acinetobacter baumannii* and increased susceptibility of these strains to beta-lactams [19]. The data for antibacterial activity of combinations between NAC and tetracycline antibiotics are controversial. NAC modulated the MIC of tetracycline depending on the strain [39]. Doxycycline in combination with NAC insignificantly stimulated the growth of *T. pyogenes*, cultured under conditions for biofilm formation [39]. Additional studies are required to evaluate the effect of doxycycline and NAC combination against *M. gallisepticum* before further recommendations about their simultaneous use.

In vitro Pk-Pd models for *M. gallisepticum*, based on dynamic time-killing curves, were used to characterize the effect of doxycycline. The estimated %*T* > MIC value for 3log10 (CFU/mL) reduction was 54.36% during the 48 h treatment period [14]. Time-dependent activity of the antibiotic against *M. gallisepticum* was determined, and %*T* > MIC was proposed as a reliable Pk-Pd index [14]. Correction for protein binding is crucial for correct prediction of the efficacy of antibiotics. Protein binding of doxycycline in broilers with natural mycoplasmosis is reported to be 56%, which is similar to our findings for a concentration range between 0.1 and 1 mg × L^−1^; therefore, the value of *f* was fixed to 0.5 [11]. A Pk-Pd index %*f*T > MIC ≥ 50 is often applied for antibiotics with time-dependent activity, including tetracyclines [14,40,41]. However, in the study, conducted by Zhang et al. [14], regrowth of *M. gallisepticum* was detected promptly after a drop of the concentrations below MIC, which indicates that a higher Pk-Pd target could be required for higher efficacy. The PTA of the registered dose of doxycycline for chickens (20 mg/kg b.w./24 h) was evaluated taking into account the Pk-Pd index %*f*T > MIC ≥ 80 [14,41,42]. Based on the proposed Pk-Pd index ≥ 80%, the registered dose of doxycycline, administered via drinking water, was predicted to be effective against *Mycoplasma* spp. with MIC ≤ 0.5 mg × L^−1^ in 90% of the population. %*f*T > MIC of 100% can be achieved for strains with MIC of 0.25 mg × L^−1^. So far, VetCAST and CLSI did not propose a clinical breakpoint for *Mycoplasma* spp. Literature data reported MIC values between 0.16 and 0.5 mg × L^−1^ for 78 *M. gallisepticum* isolates obtained from chickens and collected from European countries [43]. The same study found four isolates with a higher MIC value of 1 mg × L^−1^. The MIC value of 2 mg × L^−1^ was registered in another study [44]. These data confirm that doxycycline can be effective against *M. gallisepticum* and also indicate that it is advisable to use the tetracycline antibiotic in practice after the antimicrobial susceptibility test.

Despite some limitations of the current study, its results can be used as a basis for further improvement of the population pharmacokinetic model by addition of more data. The bias can be further avoided if data from intravenous administration are added and if more results for absorption phase of doxycycline are available. Drug–drug pharmacokinetic interactions between doxycycline and NAC were not revealed, but the available data are not sufficient to conclude that these drugs can be co-administered for treatment of poultry if there is no information about the effect of the combination on the susceptibility of *M. gallisepticum.* The limitations of the current model can be avoided with the application of more advanced models with the implementation of experiments for the determination of time-killing curves of *M. gallisepticum* when doxycycline is applied in combination with NAC. More reliable results for the Pk-Pd breakpoint can be proposed with the PopPk model if ECOFF values for MIC are available.

## 5. Conclusions

The developed population pharmacokinetic model did not reveal a significant effect of *M. gallisepticum* infection or NAC co-administration on the pharmacokinetic parameters of orally administered doxycycline. The registered doxycycline dose of 20 mg/kg b.w. administered every 24 h in chickens produced a 90% PTA for a MIC value of 0.5 mg × L^−1^ if the target, corrected for protein binding, *f*T > MIC was set at 80%, or for a MIC value of 0.25 mg × L^−1^ if *f*T > MIC is 100%. The obtained results indicate that doxycycline can be applied for treatment of mycoplasmosis when MIC values of pathogenic strains are ≤0.5 mg × L^−1^. However, a more accurate Pk-Pd target for *M. gallisepticum* can be proposed if the ECOFF value is available.

## Figures and Tables

**Figure 1 pharmaceutics-14-02440-f001:**
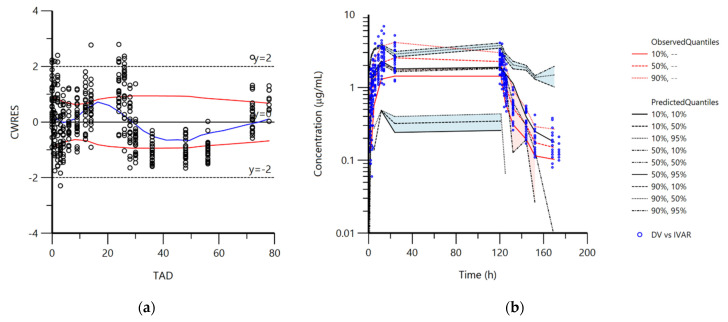
(**a**): Plot of conditional weighted residual values (CWRES) versus time after dose (TAD, h); <20% of values (3.46%) were outside the accepted limits y = 2 and y = −2, demonstrating adequate description of changes in concentrations by the pharmacokinetic model. (**b**): Visual predictive check (VPC) plot depicting the observed quantiles (10, 50, and 90%) and corresponding predicted quantiles for doxycycline concentrations.

**Figure 2 pharmaceutics-14-02440-f002:**
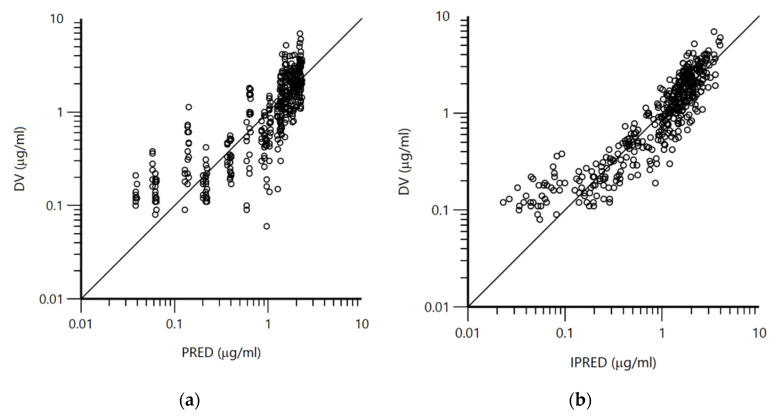
(**a**): Logarithmic plot of the dependent variable (DV, measured plasma concentrations in µg × mL^−1^) versus predicted plasma concentrations (PRED, µg × mL^−1^) of doxycycline in all treated chickens (n = 32). (**b**): Logarithmic plot of the dependent variable (DV, measured plasma concentrations in µg × mL^−1^) versus individual predicted doxycycline concentrations (IPRED, µg × mL^−1^) of doxycycline in all treated chickens (n = 32).

**Table 1 pharmaceutics-14-02440-t001:** Population parameters of orally administered doxycycline via drinking water at the dose rate of 20 mg/kg b.w./24 h for 5 days, alone or with co-administration of N-acetylcysteine via feed at a dose rate of 100 mg/kg b.w./24 h for 5 days, in healthy and in *M. gallisepticum*-infected broiler chickens (n = 32).

Parameters	Estimates	Units	SE	*CV*%	2.5% CI	97.5% CI
*tvk_a_*	0.093	1/h	0.001	1.47	0.096	0.096
*tvV/F*	4.73	L/kg	0.139	2.94	4.46	5.00
*tvk_e_*	0.154	1/h	0.002	1.16	0.15	0.16
*tvk* _12_	4.59	1/h	0.131	2.84	4.34	4.85
*tvk* _21_	7.06	1/h	0.170	2.41	6.73	7.34
tvtlag	0.80	h	0.029	3.58	0.75	0.86
tvCMultStdev	0.388	%	0.013	3.37	0.36	0.41
stdev0	0.121	µg/ml	0.006	4.94	0.11	0.13

Typical value (tv) of *k_a_*—absorption rate constant; *V*—volume of distribution; *k_e_*—elimination rate constant; *k*_12_ and *k*_21_—distribution rate constants, which characterize the distribution from the central to the peripheral compartment and vice versa; tlag—lag time; tvCMultStdev—standard deviation for multiplicative residual; and stdev0—standard deviation for additive residual error.

**Table 2 pharmaceutics-14-02440-t002:** Random effects of doxycycline after oral administration via drinking water at the dose rate of 20 mg/kg b.w./24 h for 5 days, alone or co-administered with N-acetylcysteine via feed at a dose rate of 100 mg/kg b.w./24 h for 5 days, in healthy and in *M. gallisepticum*-infected broiler chickens (n = 32).

Omega	Variance	SE	BSV (*CV*%)	Shrinkage
η*k_a_*	0.002	0.0001	0.20	0.67
η*V*	0.06	0.002	6.40	0.12
η*k_e_*	1.02 × 10^−7^	5.1 × 10^−8^	0.0001	0.33
η*k*_12_	0.32	0.012	32.82	0.22
η*k*_21_	0.14	0.0007	14.27	0.21
ηtlag	0.79	0.0008	93.17	0.10

Variance of η*k_a_*, η*V*, η*k_e_*, η*k*_12_, η*k*_21_ and ηtlag are random components of the model (eta), and BSV is the between-subject variability, calculated with Equation (3).

**Table 3 pharmaceutics-14-02440-t003:** Probability of target attainment (%) for %*f*T > MIC ≥ 80 and 100 for possible MICs of doxycycline in the range from 0.25 to 2 mg × L^−1^ and for a dosing regimen of 20 mg/kg b.w./24 h.

Dose 20 mg/kg b.w./24 h	MIC mg × L^−1^	Quantiles (%)						
		5	10	25	50	75	90	95
Time above MIC (%)	0.25	100	100	100	100	100	**100**	100
	0.5	100	100	100	100	92.65	**79.60**	72.46
	1	100	91.89	76.23	62.07	49.61	40.23	35.39
	2	44.74	38.05	27.66	17.50	9.45	4.35	2.21

## Data Availability

Data are contained within the article and Appendix A.

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
