# Peer review of "Population Pharmacokinetics of Doxycycline, Administered Alone or with N-Acetylcysteine, in Chickens with Experimental *Mycoplasma gallisepticum* Infection"

_pharmaceutics, 2022, doi:10.3390/pharmaceutics14112440_

Round 1

Reviewer 1 Report

Dear Editors and authors:

The authors determined the population pharmacokinetics of doxycycline in chickens with or without co-administration of N-acetylcysteine. And they published half of the results about N-acetylcysteine in the journal of Vet. Sci. (20218(11), 244; https://doi.org/10.3390/vetsci8110244), this half of the results are about the population pharmacokinetic of doxycycline. Based on the criteria of the Journal of pharmaceutics, I decided to Reconsider after major revision to this manuscript.

Major concern /comments:

1.     Please show the results of experimental infection with M. gallisepticum in chickens, e.g. the infection rate, and the pathological examination report. This is very important for your model and conclusion.

2.     Please revised the introduction part, and justified the reason for the co-administration of doxycycline with N-acetylcysteine.

3.     Please calculate the pharmacokinetic data obtained for all 4 groups. Was there a difference in the Tax, Cmax, and AUC between the 4 groups? Because the authors showed that Co-variables such as disease and co-administration of NAC did not improve the prediction of the population data. What is the reason? the Co-variables didn’t affect the pharmacokinetic parameters of the doxycycline? Or high BSV?

4.     Please justified the reason for selecting A two-compartmental model and the Monte Carlo Simulations.  Discussion of the limitation of the models.

5.     Did the co-administration of N-acetylcysteine improve the Probability of Target Attainment (PTA) of doxycycline? Please clarify.

6.  Pages 2 and 3, Delete the similar description of the Methods and cited your published paper.

Author Response

Comments and Suggestions for Authors

Reviewer 1

Dear Editors and authors:

The authors determined the population pharmacokinetics of doxycycline in chickens with or without co-administration of N-acetylcysteine. And they published half of the results about N-acetylcysteine in the journal of Vet. Sci. (2021, 8(11), 244; https://doi.org/10.3390/vetsci8110244), this half of the results are about the population pharmacokinetic of doxycycline. Based on the criteria of the Journal of pharmaceutics, I decided to Reconsider after major revision to this manuscript.

Answer: We hope that the Reviewer will accept our decision to divide the pharmacokinetic analysis of both compounds in two papers because we believe that the presentation of all the data in one manuscript would make it difficult for reading. Additionally, due to different mode of action of both compounds we decided to separate the data in two papers.

 Major concern /comments:

  1. Please show the results of experimental infection with M. gallisepticum in chickens, e.g. the infection rate, and the pathological examination report. This is very important for your model and conclusion.

Answer: The morbidity rate was 44.3±6.4%. The results from the pathology were available, including some pictures, but due to the fact that the disease was well described in the scientific literature we did not include additional details and pictures in the manuscript.

The pictures and the conclusion by the pathologist also proved the availability of the infection, provoked by Mycoplasma gallisepticum. The main signs of the provoked infection in chickens were as follows: unilateral aerosacculitis (right or left air sac) or bilateral aerosacculitis with thickened air sacs and serofibrinous exudate, ascites. The thoracic and abdominal air sacs were affected. In some chickens fibrinous polyserositis was observed. This information was included in the manuscript.

 Please revised the introduction part, and justified the reason for the co-administration of doxycycline with N-acetylcysteine.

Answer: The introduction part has been revised. Additional information has been added. NAC has been selected as a drug that can decrease biofilm production and to decrease the MIC values of antibiotics. Increased activity of tetracycline by NAC co-administration was determined against bovine mastitis pathogens (Yang et al., J. Dairy. Sci. 2016, 99, 4300–4302. doi: 10.3168/jds.2015-10756.). These data, the beneficial effect of NAC in the inflammation of respiratory system and the absence of literature for pharmacokinetic interactions provoke us to perform these experiments. Additional explanation has been added on lines 73-82: “NAC reduces the values of MIC of tetracycline for strains of Staph. aureus, Strep. dysgalactiae and Strep. agalactiae, except that of E. coli [20]. It is often used in the treatment of chronic bronchitis because of its mucolytic activity and antioxidant properties [21]. A literature review suggest that NAC can be an important modulator of antibiotic activity and can have beneficial effect in inflammation of the respiratory system but it does not provide information for pharmacokinetic interactions between doxycycline and NAC. Therefore, NAC was co-administered with doxycycline in chickens with respiratory infection, caused by M. gallisepticum, and its effect on pharmacokinetics of the antibiotic was examined”.

  1. Please calculate the pharmacokinetic data obtained for all 4 groups. Was there a difference in the Tax, Cmax, and AUC between the 4 groups? Because the authors showed that Co-variables such as disease and co-administration of NAC did not improve the prediction of the population data. What is the reason? the Co-variables didn’t affect the pharmacokinetic parameters of the doxycycline? Or high BSV?

Answer: We included the calculations from non-compartmental analysis in the supplementary file, Table S4. We hope that the Reviewer will accept the presentation of these data in the supplementary file because we think that the whole study resulted in a lot of data and we do not like to make the manuscript more complicated with additional type of analysis. Nevertheless, we included the minimum necessary information in the manuscript (lines 209-218; 295-299 and 388-390).

We hypothesized that the pharmacokinetics of doxycycline can be affected by the infection of Mycoplasma gallisepticum but the population model did not show improvement of the Log Likelihood (-2LL), the Akaike Information Criterion (AIC), visual inspection of the plots and plots of visual predictive check (VPC). LRT test confirm this observation. Absence of a significant changes in the population pharmacokinetic parameters can be attributed to the chronification of mycoplasmosis. It indicates that diseased condition was not associated with effect on the pharmacokinetics and therefore it cannot improve the model and BSV. Similar explanation can be given for NAC: absence of a significant effect on the pharmacokinetics of doxycycline. At the end of the manuscript a paragraph with the limitations of the study was included and the suggestion for the further improvement of the model were made.

  1. Please justified the reason for selecting A two-compartmental model and the Monte Carlo Simulations. Discussion of the limitation of the models.

Answer: A two-compartmental model fits the best to the data. The reason for selection of a two-compartmental model was based on the concepts of population modelling: “The Akaike information criterion (AIC) and Bayesian information criterion (BIC or Schwarz criterion) are useful for comparing structural models” (Mould & Upton, CPT: Pharmacomet. Syst. Pharmacol. 2012, 1, e6. doi: 10.1038/psp.2012.4.; Mould & Upton, CPT: Pharmacomet. Syst. Pharmacol. 2013, 2, e38. doi: 10.1038/psp.2013.14.). Additionally, we used the model comparer tool of the pharmacokinetic software Phoenix. Several models were checked and the proposed model in our view was selected as the most appropriate. As described in the text, the final population model was selected on the basis of the lowest values of Log Likelihood (-2LL), the Akaike Information Criterion (AIC), visual inspection of the plots, plots of visual predictive check (VPC) and model comparer tool of Phoenix. The text has been clarified by adding “model comparer tool of Phoenix”. The cited tool and the LRT test were applied as has been described on lines 242-243.

Monte-Carlo simulations were used to generate the time-concentrations curves for a bigger population. According to the cited articles in the manuscript and to Trang et al. (2017) the Monte Carlo simulation is used to generate data for pharmacokinetic–pharmacodynamic (PK-PD) target attainment analyses to assess antibacterial dosing regimens in early and late stage drug development (Trang et al., Curr Opin Pharmacol. 2017;36:107-113. doi: 10.1016/j.coph.2017.09.009.). On lines 264-266 we explained how and why Monte Carlo Simulations has been used: “The established population pharmacokinetic model was used as a basis for generation of a virtual population by Monte Carlo Simulations in Phoenix 8.3.4. In total 5000 Monte Carlo simulations were conducted to calculate estimates of % fT > MIC.” In general, at the moment these so called “in silico” tools are accepted by EMA as reliable models for evaluation of efficacy and for avoiding of the risk for non-prudent use of antibiotics.

Some limitations of the model were pointed in the last paragraph of the discussion. We tried to clarify additionally the limitations by adding the following sentence within the last paragraph: “The limitations of the current model can be avoided with application of more advanced models with implementation of experiments for determination of time-killing curves of M. gallisepticum, when doxycycline is applied in combination with NAC.”

  1. Did the co-administration of N-acetylcysteine improve the Probability of Target Attainment (PTA) of doxycycline? Please clarify.

Answer: With application of PTA analysis, we calculated the percentage of chicken in which the PK/PD target is met (the PTA value) at a given MIC. The co-administration of NAC did not influence the pharmacokinetics of doxycycline in chickens. Our previous paper demonstrated that NAC can be found at very low concentrations in plasma. Our unpublished data revealed that at these low concentrations NAC does not change MIC values of doxycycline against Gram positive and Gram negative strains of microorganisms. Therefore, we did not expect effect on PTA after co-administration of NAC. The final model did not simulate separately time-concentration curves for chickens treated with doxycycline alone or in combination with NAC, respectively. Probability of Target Attainment analysis was done on the basis of simulations of data for bigger population and they were performed after building of the population model and with means of the population analysis. We relied on the assumption that the population pharmacokinetic analysis improves its power and reliability with increasing of the number of the included subject (proved in papers such as Ribbing, J., Jonsson, E., J Pharmacokinet Pharmacodyn 31, 109–134 (2004). https://doi.org/10.1023/B:JOPA.0000034404.86036.72; Duffull et al., Br J Clin Pharmacol. 2011;71(6):807-14. doi: 10.1111/j.1365-2125.2010.03891.x.; Temmerman et al., Antibiotics 2021, 10, 604. https:// doi.org/10.3390/antibiotics10050604).

One of the limitations of the study was that we used published data for MIC of Mycoplasma spp. and this is explained in the last paragraph of the manuscript.

  1. Pages 2 and 3, Delete the similar description of the Methods and cited your published paper.

Answer: Only essential information for the understanding of the experimental design has been left in the manuscript as well as the information which was not included in the published article.

Reviewer 2 Report

Line 2-3:

It should be include in the title of publication that the effect of NAC on the pharmacokinetic of doxycycline has been investigated.

Line 15:

Replace „gallisepticum”

Line 16, 17, 340:

Missing the b.w.

Line 98:

It is useful putting the title on the next page. „Experimental design”

Line 150:

It is useful putting the paragraph on the next page. „Blood samples (0.8 ml)…”

Line 154:

Replace rpm by g

Line 200:

It is useful putting the title on the next page. „Population pharmacokinetic analysis”

Line 253:

Replace Pharmacokinetic-pharmacodynamic

Line 111-118 and 263-268:

In my opinion, it would have been worthwhile to continuously measure the body weight gain, feed consumption and water consumption of the infected and uninfected groups and to compare these indicators to confirm the successful Mycoplasma infection of the infected group.

In addition, a PCR test should be performed to demonstrate the success of the infection and the freedom of the non-infected group.

Furthermore, it would have been useful to compare feed consumption, daily weight gain and water consumption of the groups consuming feed supplemented with NAC with those of the groups not consuming NAC, in order to demonstrate that acetylcysteine does not affect these indicators.

The assessment is well presented and the interpretation of the results is appropriate. The discussion section and the conclusion are sufficiently detailed and understandable. The topic is extremely important and research in this direction should be encouraged. The work is a niche for both science and veterinary practice.

Author Response

Comments and Suggestions for Authors

Reviewer 2

Line 2-3:

It should be include in the title of publication that the effect of NAC on the pharmacokinetic of doxycycline has been investigated.

Answer: A new title was selected for the revised version: “Population pharmacokinetics of doxycycline, administered alone or with N-acetylcysteine, in chickens with experimental Mycoplasma gallisepticum infection”. We hope that now, it reflects better the content.

Line 15:

Replace „gallisepticum”

Answer: We are thankful for this remark. The change was performed.

Line 16, 17, 340:

Missing the b.w.

Answer: Again, thank you for this remark. The revision has been made.

Line 98:

It is useful putting the title on the next page. „Experimental design”

Answer: After the performed changes in the title, „Experimental design” moved to the next page where is its more proper place.

Line 150:

It is useful putting the paragraph on the next page. „Blood samples (0.8 ml)…”

Answer: The answer is similar to the previous one: the paragraph moved to the next page where is its more proper place.

Line 154:

Replace rpm by g

Answer: Thank you for this remark. The value of g was included in the text. With the used centrifuge 3000 rpm is equal to 850 g.

Line 200:

It is useful putting the title on the next page. „Population pharmacokinetic analysis”

Answer: Again, the answer is similar to the previous remarks for the place of subheadings.

Line 253:

Replace Pharmacokinetic-pharmacodynamic

Answer: The full term “Pharmacokinetic-pharmacodynamic” was removed and the abbreviation was used.

Line 111-118 and 263-268:

In my opinion, it would have been worthwhile to continuously measure the body weight gain, feed consumption and water consumption of the infected and uninfected groups and to compare these indicators to confirm the successful Mycoplasma infection of the infected group.

Answer: The body weight gain, feed consumption and water consumption has been measured during the experiment since the first day of the accommodation of the chickens at the Biobase of the Faculty of Veterinary Medicine. As explained above, we did not add this information in the manuscript because there are no new data about the disease, published for first time. They are as follow:

Because FCR reflects body weight gain related to the feed consumption we provide this parameter:

FCR healthy: 1.63±0.06

FCR infected: 2.32±0.17 (P<0.01)

Water consumption was measured during the first seven days after the infection and during the treatment. Unfortunately, we did not have additional data.

Water consumption healthy (chickens aged 1-7 days): 86.92±0.82 ml/bird/day

Water consumption infected (chickens aged 1-7 days): 45.22±2.94 ml/bird/day (P<0.01)

Water consumption healthy (chickens aged 40-45 days): 265.88±74.51 ml/bird/day

Water consumption infected (chickens aged 40-45 days): 238.21±40.81 ml/bird/day (P=0.254)

 In addition, a PCR test should be performed to demonstrate the success of the infection and the freedom of the non-infected group.

Answer: The authors are very thankful for the collaboration with the pharmaceutical company which performed another study with infection model with M. gallisepticum infected chickens at the same time. They performed all necessary ELISA test and PCR tests to prove the infection model. The hatchery was also carefully selected. The pharmaceutical company tested the infection dose before all the experiments and provided the infection suspension. The chickens for the current experiment and the experiment carried out for the needs of the pharmaceutical company were obtained at the same date, from the same hatchery and were challenged at the same date with the same suspension of M. gallisepticum.

Furthermore, it would have been useful to compare feed consumption, daily weight gain and water consumption of the groups consuming feed supplemented with NAC with those of the groups not consuming NAC, in order to demonstrate that acetylcysteine does not affect these indicators.

Answer: The feed consumption and water consumption has been evaluated during the experiments because we wished to control the obtained doses. The statistical analysis was performed and the data were included in the supplementary file. There was not statistically significant difference in the water consumption between all of the groups. No differences exist in the feed consumption between the groups of healthy broilers that received doxycycline and doxycycline+NAC. Significant differences in this parameter were found between the group of healthy broilers which received doxycycline and both groups of the infected broilers. Difference was found between healthy and infected broilers treated with doxycycline+NAC and between both infected groups. These data suggest that there is no specific effect of NAC on the feed consumption. Unfortunately, we do not have data to calculate the daily weight gain because we did not expect to observe statistically significant differences for only five days of treatment and due to already existing statistically significant differences in the body weight and FCR between healthy and diseased chickens.

The assessment is well presented and the interpretation of the results is appropriate. The discussion section and the conclusion are sufficiently detailed and understandable. The topic is extremely important and research in this direction should be encouraged. The work is a niche for both science and veterinary practice.

Answer: We are thankful to the Reviewer for these comments.

Round 2

Reviewer 1 Report

All my comments were properly answered by the authors , I sugested to accept the manuscripts after minor revison of the text editing.

Reviewer 2 Report

In my opinion you should correct doxycicline to doxycycline in your supplementary file.